# Molecular Pathogenesis in Myeloid Neoplasms with Germline Predisposition

**DOI:** 10.3390/life12010046

**Published:** 2021-12-29

**Authors:** Juehua Gao, Yihua Chen, Madina Sukhanova

**Affiliations:** Department of Pathology, Northwestern University Feinberg School of Medicine, Chicago, IL 60611, USA; y-chen5@northwestern.edu (Y.C.); madina.sukhanova@nm.org (M.S.)

**Keywords:** myeloid neoplasm, germline predisposition, pathways

## Abstract

Myeloid neoplasms with germline predisposition have recently been added as distinct provisional entities in the 2017 revision of the World Health Organization’s classification of tumors of hematopoietic and lymphatic tissue. Individuals with germline predisposition have increased risk of developing myeloid neoplasms—mainly acute myeloid leukemia and myelodysplastic syndrome. Although the incidence of myeloid neoplasms with germline predisposition remains poorly defined, these cases provide unique and important insights into the biology and molecular mechanisms of myeloid neoplasms. Knowledge of the regulation of the germline genes and their interactions with other genes, proteins, and the environment, the penetrance and clinical presentation of inherited mutations, and the longitudinal dynamics during the process of disease progression offer models and tools that can further our understanding of myeloid neoplasms. This knowledge will eventually translate to improved disease sub-classification, risk assessment, and development of more effective therapy. In this review, we will use examples of these disorders to illustrate the key molecular pathways of myeloid neoplasms.

## 1. Introduction

Myeloid neoplasms unite a group of heterogeneous hematologic diseases that harbor a set of specific genetic variations, often occurring as a result of stepwise accumulation of genetic alterations in the hematopoietic stem cells or progenitor cells. Germline contributions, initially noted in families with enrichments of hematologic malignancies in first-degree relatives, eventually gained recognition as a provisional category named ‘myeloid neoplasms with germline predisposition’ in the most recent update of WHO’s Classification of Tumours of Haematopoietic and Lymphoid Tissues [1]. This category includes myeloid neoplasms with preceding cytopenias and platelet disorders (e.g., inherited variants in *GATA2* (*GATA-bindi**ng protein 2*), *RUNX1* (*runt-related transcription factor 1*), and *ETV6* (*ETS Variant Transcription Factor 6*)), myeloid neoplasms lacking other preexisting dysfunctions (e.g., inherited variants in *CEBPA* (*CCAAT enhancer binding protein alpha*) and *DDX41* (*DEAD-Box Helicase 41*)), and myeloid neoplasms in the setting of bone marrow failure syndromes (e.g., Fanconi anemia) (Table 1). There is a growing list of genes identified in recent years that may be included in this category. The incidence of germline mutations predisposing to hematologic neoplasms in the general population remains poorly defined; however, the current estimate in populations of patients with myeloid neoplasms varies significantly from 4.4% to 18% [2,3].

The study of germline mutations in myeloid neoplasms provides profound insights into leukemogenesis and a potential direction for therapeutic intervention. We do not intend to provide a comprehensive review of all of the described genes that are associated with germline predisposition to hematologic malignancies. The goal of this review is to use several specific examples of these disorders to illustrate the key molecular pathways leading to myeloid neoplasms. In the list of germline mutations with predisposition to myeloid neoplasms, many encode known master transcription factors, such as *CEBPA*, *RUNX1,* and *GATA2*. Somatic mutations in these genes are frequent in myeloid neoplasms—particularly acute myeloid leukemia (AML). Other genes that have been added more recently have more diverse and elusive biological functions. For example, *DDX41*, *SAMD9* (*Sterile Alpha Motif Domain Containing 9*), and *SAMD9L* (*Sterile Alpha Motif Domain Containing 9 Like*) have been described as being involved in innate immunity, antiviral responses, and RNA processing and endocytosis. These genes may uncover new pathways that confer genetic predisposition. Aberrant activation of important pathways is a common finding in hematologic malignancies. Therefore, gain-of-function germline mutations in genes, such as *NRAS* (*NRAS Proto-Oncogene*, *GTPase*), *KRAS* (*KRAS Proto-Oncogene*, *GTPase*), *CBL* (*Cbl Proto-Oncogene*), *PTPN1* (*Protein Tyrosine Phosphatase*
*Non-Receptor Type 11*), and *SH2B3* (*SH2B Adaptor Protein 3*), result in aberrant activation of corresponding signaling pathways, resulting in myeloid neoplasms, such as Juvenile myelomonocytic leukemia (JMML) or myeloproliferative neoplasm (MPN). Predisposition to myeloid neoplasms can also occur in the setting of inherited disorders, such as bone marrow failure syndromes (i.e., Fanconi anemia, Diamond–Blackfan anemia, Dyskeratosis congenita, etc.), constitutional mismatch repair deficiency syndrome, or Li–Fraumeni syndrome due to germline *TP53* mutations.

## 2. Transcription Control

Transcription factors recognize and bind to their specific consensus sequence elements and regulate downstream gene expression, which is critical for normal hematopoiesis. The dysregulation of transcription factors caused by gene mutations, chromosomal aberrations, or aberrant expression is an important mechanism for developing cancer, including AML. Mutations in transcription factors such as *CEBPA*, *RUNX1*, and *GATA2* can be seen in myeloid neoplasms, both de novo or with germline predisposition.

### 2.1. CEBPA

*CEBPA* encodes a key hematopoietic transcription factor, C/EBP-α, which is involved in lineage-specific myeloid differentiation. Mutations in *CEBPA* occur in 5–10% of AML. AML with biallelic *CEBPA* mutations accounts for 2–15% of de novo AML and is a provisional entity in the WHO Classification of Tumours of Haematopoietic and Lymphoid tissues. Patients diagnosed with AML with biallelic *CEBPA* gene mutations have a longer overall survival and event-free survival compared to those with a monoallelic *CEBPA* mutation or *CEBPA*-wild-type, therefore justifying the recognition of this group as a distinct provisional entity. Normally, C/EBP-α has a full-length 42 kDa (p42) isoform. The biallelic combination of mutations is defined when one N-terminal mutation and one C-terminal mutation, which either block C/EBP-α dimerization or block DNA binding, are both present. Almost all familial cases of *CEBPA*-mutated AML have biallelic mutations, with the germline *CEBPA* mutation typically being a stop-gain frameshift variants clustered in the N terminus, resulting in a truncated 30 kDa (p30) isoform. The second mutation is typically acquired and is either a missense variant or in-frame insertion or deletion occurring in the C-terminal bZIP region and resulting in abolished DNA binding or dimerization. Recent studies revealed that in addition to sequence variants, *CEBPA* promoter methylation also results in silencing of the gene; thus, epigenetic control of expression of genes crucial for hematopoiesis may also contribute to the pathogenesis of AML or MDS [4,5]. Another interesting finding in AML patients with germline *CEBPA* mutations is that the genetic profile of the clone at the time of relapse is often distinct from the one at the time of diagnosis, including different somatic *CEBPA* mutations [6]. Tawana et al. reported a cumulative incidence of relapse in familial AML with biallelic *CEBPA* at the rate of 56% by 10 years, with some patients having more than three relapse occurrences over a period of 17 to 20 years [6]. This discovery highlights a unique pattern of disease development and progression in familial cases of AML with bi-allelic *CEBPA*. The most common co-occurring mutations in AML with biallelic *CEBPA* mutations are variants in *GATA2, WT1* (*WT1 transcription factor*), *TET2* (*Tet Methylcytosine Dioxygenase 2*), and *CSF3R* (*Colony Stimulating Factor 3 Receptor*). Other genes commonly mutated in myeloid neoplasms, such as *FLT3* (*fms like tyrosine kinase 3*), *DNMT3A* (*DNA Methyltransferase 3 Alpha*), *IDH1/2* (*Isocitrate Dehydrogenase* (*NADP*(*+*)) *1/2*), *NPM1* (*Nucleophosmin 1*), and *RUNX1*, are rare and almost mutually exclusive [7]. Although limited, the available literature reported high penetrance of germline *CEBPA* mutations, leading to almost 100% lifetime risk of developing subsequent AML [6], in contrast to other genes with frequent heritable mutations, such as *RUNX1*, which confers almost ~35% lifetime risk of myeloid neoplasm.

### 2.2. RUNX1

*RUNX1* encodes the DNA-binding subunit of the core-binding factor complex and a master–regulator transcription factor (TF) involved in hematopoiesis. *RUNX1* mutations occur in ~10% of adult AML cases. Although most *RUNX1* mutations in AML are acquired, germline *RUNX1* variants are not rare and represent 8–10% of *RUNX1*-mutated AML cases [8,9,10,11]. In contrast to AML with germline mutations in *CEBPA*, which often do not have precedent hematologic abnormalities, individuals with pathogenic germline *RUNX1* variants may present as autosomal-dominant familial platelet disorder (FPD) with a high propensity for development of hematologic malignancies. Approximately 44% of individuals with *RUNX1*-FPD develop AML or MDS with a median age of onset of 33 years [12,13]. Germline mutations in *RUNX1* encompass a whole spectrum ranging from missense, stop-gain, and frameshift variants to partial and whole gene deletions. Most missense *RUNX1* germline variants cluster in the RUNT domain. However, missense mutations outside of the RUNT domain—in particular, in consensus splicing sites or exon-level deletions—have also been reported [8]. The mechanisms of pathogenicity of different *RUNX1* mutations may be different; thus, missense variants in the RUNT domain may cause loss of binding activity, whereas stop-gain or frameshift variants or whole gene deletions result in protein loss-of-function with haploinsufficiency. In patients with germline *RUNX1* mutations, a second allele often acquires somatic mutation at the time of progression to an overt myeloid neoplasm, likely as a result of an impaired DNA repair pathway and an increased mutagenicity. Interestingly, unlike in the *CEBPA* example, biallelic *RUNX1* mutations are rarely seen in sporadic AML cases. Other co-mutations often seen in myeloid neoplasms with germline *RUNX1* are in *DNMT3A*, *FLT3*, *GATA2*, *PHF6* (*PHD finger protein 6*), *BCOR* (*BCL6 Corepressor*), *WT1*, and *TET2* [14,15,16].

### 2.3. GATA2

*GATA2* encodes a transcription factor with several functionally important domains, including the N-terminal and C-terminal zinc finger (ZF1 and ZF2) domains, a nuclear localization signal, and poorly defined transactivation domains. Gata2 binds to the consensus sequence W/GATA/R (W = A or T and R = A or G) in the promoter/enhancer regions of many target genes, such as *PU.1*, *LMO2*, *TAL1*, *FLI1,* and *RUNX1*, which regulate self-renewal of hematopoietic stem cells and promote differentiation to more mature blood cells. It is also involved in the development of the lymphatic system. If this gene is mutated, the disease penetrance is high. It was estimated that individuals with germline *GATA2* mutations have almost 90% risk of developing clinical symptoms and 50–70% risk of developing myeloid neoplasms [17,18]. Pathogenic *GATA2* germline variants often occur in the exonic and intronic/regulatory regions of the gene and include deletions, missense, nonsense, frameshift, and splice-site changes and alterations in intronic regulatory elements. Most pathogenic variants lead to haploinsufficiency due to an inability of non-functional proteins to bind DNA or other transcription factor partners. Acquisition of *ASXL1* mutations may cooperate with germline *GATA2* mutations as a driver of leukemogenesis [19]. Other concurrent mutations have been reported in genes, including *RUNX1*, *TP53*, *STAG2*, *SETBP1* (*SET Binding Protein 1*), *CBL*, *EZH2* (*Enhancer of Zeste 2 Polycomb Repressive Complex 2 Subunit*), *NRAS/KRAS*, *JAK3* (*Janus Kinase 3*), and *PTPN11* [20]. At the same time, *SF3B1* (*Splicing Factor 3b Subunit 1*), *U2AF1* (*U2 Small Nuclear RNA Auxiliary Factor 1*), *NPM1*, and *FLT3* are rarely mutated in *GATA2*-mutated myeloid neoplasms [21,22].

## 3. Splicing and Signal Transduction Control

The dysregulation of alternative splicing has emerged as an important mechanism for myeloid neoplasms. Recent studies have shown that alternative pre-mRNA splicing is common in myeloid neoplasms, although the functional relevance of the splicing differences remains elusive. Most of the mutations in splicing factors are somatic; however, rare germline variants, such as *DDX41* mutations, have recently been added to the list of contributors to germline predisposition. Another new addition to the list is *SAMD9*/*SAMD9L*, which is involved in endocytosis, growth factor signaling, and the antiviral inflammatory process. The functional mechanism in the pathogenesis of myeloid neoplasms is largely uncharacterized and is an area for active research. In contrast, it is well established that dysregulated cytokine and growth factor signaling pathways are common in myeloid leukemia. Germline variants involving genes along the *RAS*/*MAPK* pathway are associated with a diverse group of congenital disorders with high risk of development of myeloproliferative neoplasms—most notably, juvenile myelomonocytic leukemia. These entities are grouped as “juvenile myelomonocytic leukemia associated with neurofibromatosis, Noonan syndrome, or Noonan syndrome-like disorders” in the most recent update of the WHO Classification of Tumours of Haematopoietic and Lymphoid Tissues.

### 3.1. DDX41

Germline *DDX41* mutations have recently been added to the list of genes involved in familial MDS and AML. *DDX41* (DEAD-Box Helicase 41) is located on chromosome 5q35 and encodes an RNA helicase protein with a function in RNA splicing. Mutations of spliceosomes are common in myeloid neoplasms, but are generally mutually exclusive with *DDX41* mutations [23]. Deep total RNA sequencing revealed that *DDX41* defects are associated with either more avid exon skipping or more exon retention [23]. Reports of germline *DDX41* mutations have indicated specific mutational hot-spots with ethnic associations; thus, p.M1I and p.D140Gfs*2 germline variants are enriched in Caucasian populations, while the p.A500Cfs*9 germline change has been reported in families of Asian descent [23,24]. Acquisition of the p.R525H variant in *DDX41* was reported as the most frequent somatic event in tumors. Germline *DDX41* mutations strongly predispose individuals to an acquired mutation of the second *DDX41* allele. Myeloid neoplasms with germline *DDX41* mutations are frequently present with hypocellular bone marrow, erythroid dysplasia, and high-risk MDS or AML. Quesada et al. reported that approximately 60% of mutant *DDX41*-driven AML arose from antecedent MDS [25]. The additional concurrent somatic mutations in *ASXL1* (*ASXL Transcriptional Reg**ulator 1*), *EZH2, SRSF2* (*Serine And Arginine Rich Splicing Factor 2*), *CUX1* (*Cut like homeobox 1*), and *SETBP1* have also been reported as being strongly associated with secondary AML rather than *de novo* AML [26]. *DDX41* defects led to loss of tumor suppressor function due to altered pre-mRNA splicing and RNA processing. Germline *DDX41* mutations reportedly represented 2.4% of a large cohort of 1385 unselected adult patients with MDS and AML [27]. In contrast to other genes with strong predisposition to familial myeloid neoplasms, such as *GATA2*, *RUNX1*, or *CEBPA*, which are frequently mutated in sporadic MDS/AML cases, somatic *DDX41* variants are exceedingly rare (0.4%) in the absence of predisposing germline *DDX41* variants and were reported only in five out of 1385 unselected patients with MDS or AML [27]. In addition, quite differently from other genes with germline predisposition to myeloid neoplasms with a younger age of disease onset, the median age at the time of diagnosis of hematologic malignancy for *DDX41-*mutated cases was 69 years, which is not significantly different from that of sporadic MDS/AML [27].

### 3.2. SMAD9 and SMAD9L

*SMAD9* and *SMAD9L* are two interferon-inducible genes located on the long arm of chromosome 7 (7q21.3). The function of the SAMD9 and SAMD9L proteins in hematopoiesis is not entirely clear, but they appear to be involved in endocytosis and cytokine signaling [28]. Monosomy 7 and interstitial deletions of 7q (-7/7q-) are well-known abnormalities that are frequently identified in MDS and AML. Haploinsufficiency of *SAMD9L* and/or *SAMD9* genes (both located on 7q) due to -7/7q- may contribute to development of myeloid neoplasms. Activating mutations in *SAMD9* and *SAMD9L* genes have initially been described as being associated with a clinical spectrum of disorders, including the MIRAGE (myelodysplasia, infection, restriction of growth, adrenal hypoplasia, genital phenotypes, and enteropathy) syndrome, ataxia-pancytopenia syndrome, and myelodysplasia and leukemia syndrome with monosomy 7 syndrome [29]. These germline mutations cause gain of function of *SAMD9* and *SAMD9L*, which normally suppress myeloid proliferation and inhibit cell cycle progression, leading to pancytopenias, as well as other organ hypoplasias and growth restrictions [30]. Schwatz et al. reported that germline variants of *SAMD9* or *SAMD9L* were present in 17% of pediatric MDS patients [31]. Most of these variants are missense mutations and tend to cluster in the second half of the proteins within or near a putative P-loop nucleoside triphosphatase domain. The germline nature could be obscured due to the lower variant allele frequencies depending on the extent of the -7/7q-. Interestingly, studies have shown acquired loss-of-function mutations in *SAMD9* or *SAMD9L* as a rescue from the deleterious effects of the gain-of-function germline effect. It has also been proposed that selective loss of chromosome 7, which harbors the mutant allele, occurs as a cellular adaptive mechanism to the germline *SAMD9* and *SAMD9L* variants [29,31]. This haploinsufficiency of numerous other genes located in this region, including *EZH2*, *CUX1,* and *KMT2C/MLL3* (*Lysine Methylatransferase 2C/mixed-lineage leukemia 3*), may also contribute and eventually lead to MDS and AML. These relevant or cooperating genetic alterations may influence the clinical phenotype of the individuals.

### 3.3. RAS/MAPK Pathway

The RAS superfamily of proteins includes *HRAS*, *NRAS*, and *KRAS*, which are small GTPases that play an important role in the control of various cell signaling pathways, including the MAPK pathway. RAS is active in the guanosine triphosphate (GTP)-bound state, controlling downstream signaling pathways, and is inactive in the guanosine diphosphate (GDP)-bound state. The downstream MAPK pathway has been reported as crucial for many cellular processes, such as proliferation and differentiation [32,33]. While the somatic variants in *KRAS*, *NRAS*, *PTPN1*, *NF1* (*Neurofibromin 1*), and *CBL* involved in RAS/MAPK pathway are frequent in different types of cancers, germline variants in these genes are associated with a diverse group of congenital disorders (neurofibromatosis type 1, cardiofaciocutaneous syndrome, Costello syndrome, Noonan syndrome, and Noonan syndrome with multiple lentigines), often referred to as ‘RASopathies’. These conditions are characterized by a high risk of development of myeloproliferative neoplasms—most notably, juvenile myelomonocytic leukemia (JMML) [34,35,36]. JMML is an aggressive pediatric myeloid neoplasm with a dysregulated RAS/MAPK signaling pathway due to, most frequently, either heterozygous somatic gain-of-function mutations in *KRAS*, *NRAS*, or *PTPN1* or germline RASopathy mutations in *NF1* or *CBL* tumor suppressors with subsequent biallelic inactivation in hematopoietic cells, resulting in JMML [37]. Germline mutations in *CBL* in patients with JMML are predominantly located in intron 7 and exons 8 and 9 (linker and RING finger domain); chromosome 11q isodisomy, also known as copy-neutral loss of heterozygosity (CN-LOH), was reported as the most frequent mechanism resulting in the homozygous state of *CBL* mutations with no other secondary genetic abnormalities [38,39,40]. Germline mutations in the tumor suppressor gene *NF1* are recurrent in JMML and are found in approximately 10% of patients. *NF1* encodes neurofibromin, a GTPase-activating protein and a known repressor of RAS; thus, loss of one copy of *NF1* due to germline mutation and subsequent loss of the second *NF1* allele in JMML result in RAS hyperactivity in leukemia cells. Three main mechanisms may lead to the loss of a second *NF1* in JMML: CN-LOH affecting chromosome 17q, somatic interstitial deletions, or a second *NF1* mutation [41].

## 4. Bone Marrow Failure Syndromes and Other Inherited Disorders

Myeloid neoplasms associated with bone marrow failure syndromes, telomere biology disorders, Down syndrome, and other inherited disorders are grouped in the category of “myeloid neoplasm with germline predisposition and other organ dysfunction” in the most recent update of the WHO classification. Their clinical manifestations, molecular mechanisms, diagnostic modalities, and clinical outcomes are diverse. The respective pathologic pathways involve DNA repair (Fanconi anemia), telomere biology (Dyskeratosis congenita), ribosome biogenesis (Diamond–Blackfan anemia and Shwachman–Diamond syndrome), DNA repair, and genomic instability (Li–Fraumeni syndrome). The current classification emphasizes the importance of approaching these hematologic disorders in the context of a diverse genetic and clinical spectrum.

### 4.1. Fanconi Anemia (FA)

The risk of developing a myeloid neoplasm is increased in patients with bone marrow failure syndromes, including Fanconi anemia (FA), severe congenital neutropenia, dyskeratosis congenita, Shwachman–Diamond syndrome, and Diamond–Blackfan anemia. Although the molecular mechanisms of these disorders have not entirely been elucidated, the concept of dysfunctional DNA repair being responsible for the main pathophysiology of FA is well accepted. As a result, cells from patients with FA display hypersensitivity to DNA cross-linking agents, such as mitomycin C (MMC) and diepoxybutane (DEB), revealing an increased rate of chromosome breakage upon exposure to one of these two agents. The chromosome breakage test has been developed as a clinical diagnostic test for patients with clinical suspicion of having FA. If positive, next-generation sequencing testing with a panel of FA genes is recommended to detect mutations and affected genes associated with FA for further family studies in order to identify mutation carriers. Many FA genes have been identified and grouped into broad categories: the FA core complex, ID2 complex proteins (*FANCD2* (*FA Complementation Group D2*), *FANCI* (*FA Complementation Group I*)), and a group of proteins in the downstream functional units. Proteins in the FA core complex work together to activate the ID2 complex and downstream proteins to bring in DNA repair proteins. Mutations in any of the genes involved in the FA pathway impair DNA repair, especially the homologous recombination repair of double-strand DNA damage. Hematopoietic elements are particularly sensitive to this defect. According to the International Fanconi Anemia Registry Study, the risk of developing either MDS or AML before the age of 20 is 27%, and it rapidly increases to 52% by the age of 40 [42]. The mechanism of leukemogenesis in FA is thought to be due to emerged malignant clones harboring mutations that allow them to evade cell cycle regulation and apoptosis, leading to MDS and AML [43].

### 4.2. Dyskeratosis Congenita (DC)

Another inherited bone marrow failure disorder is DC, one of a spectrum of telomere biology disorders based on shared genetic causes and telomere defects. Pathogenic variants in genes encoding core telomerase components (*TERT* (*Telomerase Reverse Transcriptase*) and *TERC* (*Telomerase RNA Component*)), or genes involved in telomere maintenance and function, have been identified in 60–70% of patients with clinical features of DC. DC is also a cancer predisposition syndrome, likely due to the genomic instability that can arise in the context of very short telomeres. Although the exact mechanism for developing myeloid neoplasms in patients with DC is still unclear, it was observed that somatic mutations in typical MDS-related genes are rare events in adult DC patients [44]. Therefore, accumulation of myeloid-associated clonal mutations does not seem to be the predominant mechanism for the initiation of myeloid neoplasms in DC patients. Other mechanisms, including the role of chromosomal instability due to critically short telomeres, may contribute to malignant transformation and expansion.

### 4.3. Diamond–Blackfan Anemia (DBA)

Similarly to FA and DC, DBA is also a cancer predisposition syndrome, although the cancer risk appears lower. DBA is the prototypic disorder linked to defects in ribosome synthesis known as ribosomopathies. The ribosomal protein genes harbor heterozygous loss-of-function mutations, resulting in haploinsufficiency. A study with a large cohort of DBA patients demonstrated an increased risk of developing MDS, AML, and solid tumors. There is accumulating evidence that ribosome dysfunction can drive malignant transformation due to the stimulation of protein synthesis. Recent studies also supported a role for p53 activation via ribosome stress signaling in leukemogenesis.

### 4.4. Li–Fraumeni Syndrome

Li–Fraumeni syndrome is a genetic disorder with a high risk of a wide range of malignancies, including hematologic neoplasms. As a classic cancer predisposition disorder, Li–Fraumeni syndrome is commonly associated with germline mutations of the p53 tumor suppressor gene, resulting in defective regulatory control over cell proliferation and homeostasis, the cell cycle, DNA repair, and genomic instability. Although hypodiploid acute lymphoblastic leukemia is the most recurrent leukemia in Li–Fraumeni patients, neoplasms of myeloid origin, including MDS and AML, are also frequent, typically developing as a therapy-related complication after the treatment of primary cancer, whereas the frequency of germline *TP53* variants in de novo AML was reported at only 1.1% [45,46].

## 5. Conclusions

Myeloid neoplasms with germline predisposition have been increasingly recognized in clinical practice. Table 2 includes genes with a recognized inherited predisposition to myeloid neoplasms, but the rapidly expanding list of genes is well beyond the ones presented in this review. Although the incidence of myeloid neoplasms with germline predisposition is still poorly defined and the molecular pathogenesis has not entirely been elucidated for many of these entities, these cases provide unique and important insights into the biology and molecular mechanisms. An oversimplified molecular mechanism of myeloid neoplasms with germline predisposition is summarized in Figure 1. We continue to gain knowledge about the regulation of the germline genetic defects and their interactions with other genes and proteins, the role of the bone marrow microenvironment, the genotype/phenotype correlations, the clinical presentations, and the longitudinal dynamics during the process of disease progression. This group of inherited hematologic conditions offers a unique model for a better understanding of the mechanism of the development of myeloid neoplasms and disease progression. This knowledge will eventually translate into improved sub-classification, risk assessment, and development of effective therapies beyond standard options.

## Figures and Tables

**Figure 1 life-12-00046-f001:**
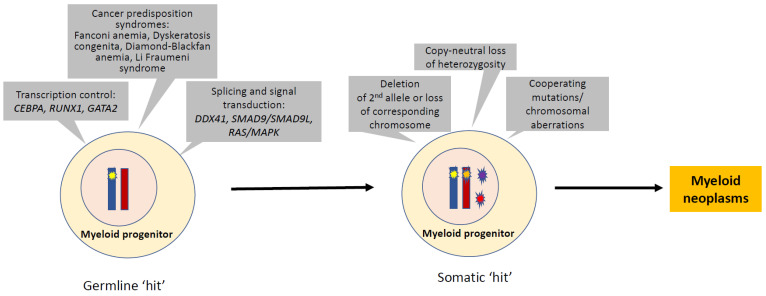
Molecular pathogenesis in myeloid neoplasms with germline predisposition. The pathogenesis of myeloid neoplasms with germline predisposition serves as a model for multi-step leukemogenesis of MDS or AML. The germline aberrations in transcription control, splicing and signal transduction, bone marrow failure, or other inherited disorders often provide the first hit. Many somatically acquired secondary events may promote a transformation that leads to overt MDS and AML. These secondary events include deletion/loss of the second allele, copy-neutral loss of heterozygosity, and acquired cooperative mutations or chromosomal aberrations.

**Table 1 life-12-00046-t001:** WHO classification of myeloid neoplasms with germline predisposition [1].

MN with germline predisposition without a preexisting disorder or organ dysfunction	AML with germline *CEBPA* mutation
	MN with germline *DDX41* mutation
MN with germline predisposition and preexisting platelet disorder	MN with germline *RUNX1* mutation
	MN with germline *ANKRD26* mutation
	MN with germline *ETV6* mutation
MN with germline predisposition and other rgan dysfunction	MN with germline *GATA2* mutation
	MN associated with bone marrow failure syndromes
	MN associated with telomere biology disorders
	Juvenile myelomonocytic leukemia associated with neurofibromatosis, Noonan syndrome, or Noonan-syndrome-like disorders
	MN associated with Down syndrome

AML: acute myeloid leukemia; MN: myeloid neoplasms.

**Table 2 life-12-00046-t002:** Genes with recognized associations with familial predisposition to myeloid neoplasms.

Gene (chr. Band)	Syndrome Name	Key Clinical and Pathologic Features	Mechanism	References
*ANKRD26* (10p12.1)	Thrombocytopenia 2	Thrombocytopenia/platelet dysfunction No organ dysfunction MDS, AML, rarely CML, CMML	*ANKRD26* germline missense mutations result in increased *MPL* signaling pathway and impaired pro-platelet formation by megakaryocytes; *ANKRD26* promoter mutations prevent *RUNX1* binding to the promoter, resulting in consecutive activation of MAPK signaling and abnormal platelet function.	WHO 2017 NCCN 2021 [47,48]
*ATG2B* (14q32.2) *GSKIP* (14q32.2)	MNs with germline predisposition due to duplications of *ATG2B* and *GSKIP*	ET or PMF progress to AML, CMML, CML, aCML Penetrance > 80%	An approximately 700 kb tandem duplication results in overexpression of *ATG2B* and *GSKIP*, enhancing sensitivity to thrombopoietin; cooperates with acquired *JAK2*, *MPL*, and *CALR* mutations during MPN development.	NCCN 2021 [49,50]
*CBL* (11q23.3)	CBL syndrome	JMML, clonal macrophage/monocyte proliferation	*CBL* is a proto-oncogene that encodes a RING finger E3 ubiquitin ligase, which plays a role in tyrosine kinase signaling; the mutational spectrum is mainly missense variants or splice-site variants in the linker region and zinc-coordinating amino acids of the RING finger domain. Heterozygous germline variants are typically accompanied by copy-neutral loss of heterozygosity (CN-LOH) of the 11q23 region with the consecutive homozygous state of the *CBL* variant; thus, a high allele frequency of the identified *CBL* variant or CN-LOH at 11q23 is suggestive of CBL syndrome.	[35,51,52]
*CEBPA* (19q13.11)	Familial AML with mutated *CEBPA*	AML (normal karyotype, blasts have aberrant CD7 expression) No organ dysfunction Median age at diagnosis: 25 years ~100% penetrance	The gene encodes granulocyte differentiation factor. Germline mutations are typically N-terminal and frame-shift variants. C-terminal mutations are typically somatic and in-frame insertions, deletions, frameshift, or missense variants; they are typically unstable, and a novel clone during recurrence is not unusual. Acquired mutations in *GATA2* and *WT1* are common and mutually exclusive.	WHO 2017 NCCN 2021 [6,47,53]
*DDX41* (5q35.3)	Familial AML with mutated *DDX41*	MDS, AML, CMML, CML Normal karyotype in 80% of patients No organ dysfunction Median age at diagnosis is 62 years	DEAD/H-box helicase gene encodes an RNA helicase protein with a function in RNA splicing. Many patients have biallelic mutations (frameshift, missense, splicing). Although rare in MNs (1.5%), if detected, *DDX41* mutation is germline in 50% of cases.	WHO 2017 NCCN 2021 [23,24,47]
*ETV6* (12p13.2)	MNs with germline *ETV6* mutation/Thrombocytopenia 5	Thrombocytopenia/platelet dysfunction and no organ dysfunction ALL, MDS, AML, CMML, PV, aplastic anemia	The gene encodes a transcription factor. Mutations abrogate DNA binding, alter subcellular localization, decrease transcriptional repression in a dominant-negative effect, and impair expression of platelet-associated genes with defective maturation of megakaryocytes. Germline variants are associated with lymphoid and myeloid neoplasms.	WHO 2017 NCCN 2021 [54,55,56]
*GATA2* (3q21.3)	Familial MDS/AML with mutated *GATA2* (known as *GATA2* deficiency syndromes)	Organ dysfunction (lymphedema, hydrocele, congenital deafness, vulnerability to viral infections) 70–75% penetrance MDS, AML, common in children with MDS and monosomy 7 Dysplasia is frequent, particularly dysmegakaryopoiesis	Gata2 is a zinc-finger transcription factor that is important in the control of hematopoiesis and autoimmunity; germline mutations are loss-of-function variants. Mutations are frequent in MDS and AML cases together with *ASXL1* mutations, indicative of a possible collaborative mechanism.	WHO 2017 NCCN 2021 [18,21,57,58,59,60]
*MBD4* (3q21.3)	Familial AML with mutated *MBD4*	AML	*MBD4*, methyl-CpG binding domain 4, encodes a DNA glycosylase that removes base lesions and initiates DNA repair. *MBD4*-deficiency AML displays a 33-fold higher mutation burden, with mostly C>T in CG dinucleotide; has a propensity for *DNMT3A*-mutated clonal hematopoiesis.	[61]
*MECOM* (3q26.2)	MECOM-associated syndrome (also known as congenital amegakaryocytic thrombocytopenia and radioulnar synostosis)	Amegakaryocytic thrombocytopenia, bone marrow failure, MDS Organ dysfunction (radiosynostosis, clinodactyly, presenile hearing loss, cardiac/renal malformations)	A range of genetic variants have been observed, including gene deletions and point mutations. Mutations are clustered within the 8^th^ zinc-finger motif of the C-terminal zinc-finger domain of *EVI1*.	[62,63,64]
*RUNX1* (21q22)	Familial platelet disorder with propensity for myeloid malignancies	MDS, AML, thrombocytopenia, platelet dysfunction, CMML No organ dysfunction Median age at diagnosis is 39 years old; lifetime risk of myeloid neoplasm is ~20~65%.	The gene encodes a transcription factor with a major role in megakaryocyte maturation, differentiation, ploidization, and pro-platelet formation. Germline mutations are typically frameshift or missense all over the gene, but predominantly in the RUNT domain and transactivation domain.	WHO 2017 NCCN 2021 [15,16,47,65]
*SAMD9* (7q21.2) *SAMD9L* (7q21.2)	Congenital SAMD9/SAMD9L mutations MIRAGE syndrome/ Monosomy 7 myelodysplasia and leukemia syndrome 2/Ataxia pancytopenia syndrome	MDS, AML, pancytopenia Organ dysfunction (hematopoietic, immunologic, endocrine, genital, neurologic) Together with *GATA2*-mutated myeloid neoplasms are the most frequent oncogenic drivers in pediatric MDS (half of pediatric MDS with monosomy 7) Penetrance is not complete	*SAMD9* encodes sterile alpha motif domain-containing protein 9, which is involved in endosome fusion and plays a role in signal transduction and proliferation. *SAMD9L* is a paralog (L stands for -like) and a negative controller of proliferation. Germline mutations cause gain of function of *SAMD9* and *SAMD9L*, which normally results in suppression of myeloid proliferation and inhibits cell cycle progression, leading to pancytopenias, as well as other organ hypoplasias and growth restrictions.	[29,31,60,66,67]
*SRP72* (4q12)	Familial aplastic anemia/MDS with *SRP72* mutation	MDS, aplastic anemia, familial leukemia, bone marrow failure	SRP72 is a component of the signal recognition particle, a ribonucleoprotein complex responsible for the translocation of nascent membrane-bound and excreted proteins to the endoplasmic reticulum.	[68]

AML: acute myeloid leukemia, MDS: myelodysplastic syndrome; PV: polycythemia vera; ET: essential thrombocythemia; PMF: primary myelofibrosis; CMML: chronic myelomonocytic leukemia; CML: chronic myeloid leukemia; aCML: atypical chronic myeloid leukemia.

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
