# Peer review of "Molecular Pathogenesis in Myeloid Neoplasms with Germline Predisposition"

_life, 2021, doi:10.3390/life12010046_

Round 1
Reviewer 1 Report
The present study has reviewed the molecular pathogenesis in myeloid neoplasms with germline predisposition. This draft summarized it very concisely, thus helping the readers to catch up the main issue quickly through this review article. Mainly it is well written thus I do not have any significant issue but some minor issues to be corrected.
- In the Abstract, line 11, it was written that “Although the incidence of myeloid neoplasms with germline predisposition is very low”, while it was also mentioned in the introduction, line 37, that it is estimated to from 4.4% to 18%. 18% is not very low frequency. Thus it needs to be revised.
- In Table 2, many meaningless dots would bother the readers to read the table. For example, in key clinical and pathologic features for CBL, there is a dot, which should be removed. In the ETV6 on the column of mechanism, there is a dot in the bottom, which also is to be removed.
- The font in the reference section, it looks like Calibri font had been used, while Palatino font has been used in the main text and tables.
- It would be helpful to add some figure or cartoon to summarize the biologic function of the genes to help readers to have better understanding.
Reviewer 2 Report
This is a nice review entitled “Molecular Pathogenesis in Myeloid Neoplasms with Germline Predisposition” written by Prof Juehua Gao and colleagues.
The review is generally well written and detailed.
I have a minor comment:
- WHO classification of myeloid neoplasm with germline predisposition is correctly enlisted in Table 1. For a greater clarity to the reader, I recommend adding before each paragraph (before Transcription control, Splicing and signal transduction control…etc.) an explanatory introduction on the genes listed below and on the molecular pathway.
